

# Green inspired synthesis of zinc oxide nanoparticles using *Silybum marianum* (milk thistle) extract and evaluation of their potential pesticidal and phytopathogens activities

Nazish Jahan[1], Kousar Rasheed[2], Khalil-Ur- Rahman[3], Abu Hazafa[4], Amna Saleem[2], Saud Alamri[5], Muhammad Omer Iqbal[6] and Md Atikur Rahman[7]

[1] Department of Chemistry, Faculty of Sciences, University of Agriculture, Faisalabad, Punjab, Pakistan
[2] Department of Chemistry, Faculty of Sciences, University of Agriculture, Faisalabad, Pakistan
[3] Department of Biochemistry, Riphah International University, Faisalabad, Pakistan
[4] Department of Biochemistry, University of Agriculture, Faisalabad, Pakistan
[5] Department of Botany and Microbiology, College of Science, King Saud University, Riyadh, Saudi Arabia
[6] Key Laboratory of Marine Drugs, the Ministry of Education, School of Medicine and Pharmacy, Ocean university of China, Qingdao, China
[7] Grassland & Forages Division, National Institute of Animal Science, Rural Development Administration, Cheonan, Republic of Korea

Corresponding author
Abu Hazafa, ahazafa@unisa.it

## ABSTRACT

**Background:** The green approaches for the synthesis of nanoparticles are gaining significant importance because of their high productivity, purity, low cost, biocompatibility, and environmental friendliness.

**Methods:** The aim of the current study is the green synthesis of zinc oxide nanoparticles (ZnO-NPs) using seed extracts of *Silybum marianum*, which acts as a reducing and stabilizing agent. central composite design (CCD) of response surface methodology (RSM) optimized synthesis parameters (temperature, pH, reaction time, plant extract, and salt concentration) for controlled size, stability, and maximum yields of ZnO-NPs. Green synthesized ZnO-NPs was characterized using UV-visible spectroscopy and Zetasizer analyses.

**Results:** The Zetasizer confirmed that green synthesized ZnO-NPs were 51.80 nm in size and monodispersed in nature. The UV-visible results revealed a large band gap energy in the visible region at 360.5 nm wavelength. The bioactivities of green synthesized ZnO-NPs, including antifungal, antibacterial, and pesticidal, were also evaluated. Data analysis confirmed that these activities were concentration dependent. Bio-synthesized ZnO-NPs showed higher mortality towards *Tribolium castaneum* of about 78 ± 0.57% after 72 h observation as compared to *Sitophilus oryzae*, which only displayed 74 ± 0.57% at the same concentration and time intervals. Plant-mediated ZnO-NPs also showed high potential against pathogenic gram-positive bacteria (*Clavibacter michiganensis*), gram-negative bacteria (*Pseudomonas syringae*), and two fungal strains such as *Fusarium oxysporum*, and *Aspergillums niger* with inhibition zones of 18 ± 0.4, 25 ± 0.4, 21 ± 0.57, and 19 ± 0.4 mm, respectively.

**Conclusion:** The results of this study showed that *Silybum marianum*-based ZnO-NPs are cost-effective and efficient against crop pests.

## INTRODUCTION

The rapidly developing field of nanotechnology is considered a significant achievement in the agricultural sector for increasing pest mortality and crop production. Bacteria, weeds, insects, and fungi have an adverse effect on agricultural resources, resulting in poor product quality and yield. Therefore, chemical pesticides such as fumigants and insecticides have been commonly used to control pests worldwide. Many issues are associated with using chemical pesticides, including harmful effects on non-target species and the environment, non-biodegradable nature, pest resistance, and human health safety. As a result of these, there is a considerable demand for biopesticides that are both safe and environmentally friendly (*Melanie et al., 2022*; *Shahbaz et al., 2023*).

Bio-pesticides are organic compounds derived from naturally occurring sources such as microorganisms, fungi, and plants used to kill pests. They are preferred over synthetic pesticides because they are safe and biodegradable (*Deka et al., 2022*). Many compounds, including *Clitoria ternatea* extract, stilbenes of grape cane, oxymatrine (alkaloid compound), and olive mill oil, have recently been experienced as bio-pesticides to kill a variety of pests (*Damalas & Koutroubas, 2018*). However, due to fewer field applications, delivery issues, and lower stability, biopesticides only captured 5% ($3 billion) of the global crop protection market, failing to overtake synthetic pesticides (*Balakrishnan et al., 2019*; *Damalas & Koutroubas, 2018*). Therefore, to overcome the delivery problems of biopesticides, an advanced emerging field of nanotechnology has been introduced to improve bio-pesticide stability and delivery time. Different types of metal nanoparticles (gold, copper, silver, and zinc) have been utilized in combination with biopesticides owing to this increasing solubility of less soluble active ingredients, durability, efficacy, improved dissolution rate, the slow release of active ingredients, and reduced degradation of active ingredients, controlled release of active components, better stability, enhanced absorption and bioavailability (*Narware et al., 2019*). Therefore, it is considered that nanotechnology can provide a green and competent alternative for managing insect pests in agriculture without disturbing natural harmony (*Bhattacharyya, Colombo & Sotiriou, 2016*).

Among the metal oxide nanoparticles, ZnO-NPs have gained significant importance due to being nontoxic, multifunctional, low-cost, biocompatible, and eco-friendly materials with many applications in different fields like the food industry, medicines, anti-fungal and as antimicrobial agent (*Espitia Rangel et al., 2012*), paints, sunscreens, and coatings due to high UV protection, optoelectronics, and optics (*Saha, Khlystov & Grieshop, 2018*), elimination of heavy metals from water (*Ishwarya et al., 2018*), and finally in medical fields such as nano diagnostics, nanomedicine, drug delivery, and gene delivery

(*Patwekar et al., 2022*). ZnO-based nano biopesticides have effectively controlled *T. castaneum* and S. *oryzae* with higher mortality rates. ZnO-NPs have recently been used as antibacterial agents in food packaging, textile fabrics, mouthwash, ointments, and lotions to stop microbial contamination (*Jiménez-Rosado et al., 2022*). ZnO-NPs have been synthesized using different physical and chemical approaches, including hydrothermal, co-precipitation, microwave-assisted, thermal decomposition, ultrasonic synthesis, sol-gel, and electrochemical methods. These methods are costly, toxic, require higher energy consumption, high temperature, and pressure, economic instability, long-term procedure, time-consuming, and employ poisonous solvents and chemicals that are harmful to public health and the environment (*Aldeen, Mohamed & Maaza, 2022*; *Rafique et al., 2017*). Therefore, as an alternative, green synthesis has been a preferred approach for synthesizing ZnO-NPs due to several significant characteristics, including safety, environmentally friendly protocols with nontoxic byproducts, mild reaction conditions, simple, reliable, single step, economical and the use of natural capping and reducing agents, it has attracted significantly more attention than the other contemporary techniques (both physical and chemical) (*Prasad et al., 2019*; *Velsankar et al., 2022*).

Different studies proposed that owing to the high synthesis rate, plant extracts are better nominees and more suitable for large-scale production of green nanoparticles than other organisms. Nanoparticles synthesized using plant extracts have more variations in size and shape than those synthesized from bacteria, fungi, and yeast (*Swaminathan & Sharma, 2019*). Therefore, plant-medicated green synthesis of metal nanoparticles is gaining popularity over the microorganism-mediated green synthesis methods because plants are easily accessible, safe to handle, and contain different physicochemical such as alkaloids, terpenoids, flavonoids, quinines, tannins, and phenols that act as a reducing and stabilizing agent (*Baker et al., 2013*). Green synthesis of ZnO-NPs by using *Ocimum americanum*, *Abelmoschus esculentus, Pisonia grandis, Berberis aristata, Cuminnum cyminum, Cassia alata*, *Kumar, Zare & Ghosh (2017)*, *Azedarach indica, Alestonias cholaris, Pongamia pinnata, Calotropis gigantea* leaf extract, *Mangifera indica* (*Rajeshkumar et al., 2018*) and *Aloe barbadensis* mille and other phytocompounds (terpenoids, alkaloids, flavonoids, tannins, carotenoids, and chlorophylls) have been already demonstrated, but they have limitations to get desire results (*Mahendiran et al., 2017*; *Sadiq et al., 2021*).

*Silybum Marianum* (Milk thistle) is a plant of the Carduus Marianum family (*Al-Hashem, Akbar & Morris, 2019*). Silymarin is a flavonoid and obtained from *Silybum Marianum*; it is a mixture of different types of polyphenolic compounds, including silychristin (silydianin), flavonolignans (isosilybin A, silybin A, silybin B, isosilychristin, isosilybin B, and a taxifolin (flavonoid)). The presence of phytochemicals is the probable mechanism for the green synthesis of nanoparticles. Terpenoids, flavonoids, quinones, carboxylic acids, amides, ketones, and aldehydes are phytochemicals that act as reducing and capping agents for various metal ions; silymarin is nontoxic and can be used as a safe herbal medicine. The extract of *Silybum marianum* was used for the synthesis of ZnO nanoparticles through Keto-enol Tautomerization (*Al-Hashem, Akbar & Morris, 2019*).

However, the current study aimed to synthesize the zinc oxide nanoparticles (ZnO-NPs) from the seed extract of *Silybum marianum* (Milk thistle) using the precipitation

method. Secondly, the optimization of synthesis parameters through RSM was performed. Then pesticide activity of NPs-ZnO-based biopesticides was analyzed against two pests, including *Tribolium castaneum* and *Sitophilus oryzae*. The present study also assessed the antibacterial (*Clavibacter michiganensis* and *Pseudomonas syringae*) and antifungal (*Fusarium oxysporum* and *Aspergillums niger*) activities of ZnO-NPs for better understanding.

## MATERIALS AND METHODS

### Chemicals

Zinc nitrate hexahydrate (≥98% pure), sodium hydroxide, ethanol (≥99.2% pure), NaCl, and tryptophan were purchased from Sigma-Aldrich, USA. All the studies were done in the Natural Product Laboratory of the Department of Chemistry, University of Agriculture Faisalabad.

### Preparation of seed extract of *Silybum marianum*

*Silybum marianum* seeds were collected from the local area of the University of Agriculture, Faisalabad, Pakistan. The collected seeds were washed, air-dried, ground into powdered form, and passed through a sieve to get a uniform sample. The seeds extract was prepared by adding 10 g seeds powder of *Silybum marianum* into 100 mL of distilled water (d.$H_2O$). Then the solution was kept at 80 °C on a hot plate (SCILOGEX MS-H-S) for 20 min. After heating, the solution was kept overnight at room temperature (27 ± 2 °C). Then the extract was filtered with the help of Whitman's filter article No. 1 and stored in a refrigerator at 4 °C for the future synthesis of ZnO nanoparticles.

### Optimization of parameters for the synthesis ZnO-NPs

After the successful preparation of plant extract, various synthesis parameters like temperature varied at 60 to 80 °C, pH 10–12, stirring time 40–120 min, and salt concentration of zinc nitrate from 0.06 to 0.6 M was optimized for the synthesis of novel ZnO based nano-biopesticides of minimum particle size. The experimental planned design through Response Surface Methodology (RSM) is shown in Table 1.

### Green synthesis of ZnO-NPs

For the synthesis of ZnO-NPs, different treatments given by RSM were performed at different conditions, as shown in Table 1. The synthesis of ZnO-NPs was carried out by adding 10 mL of seed extract of *Silybum marianum* into the 100 mL solution of 0.06 M zinc nitrate hexahydrate (SIGMA-ALDRICH UN-1514) and sodium hydroxide (SIGMA-ALDRICH UN-1823), keeping the solutions on a magnetic stirrer at 60 to 80 °C and pH 12. Seed extract reduced the zinc ions into zinc metal (*Devi & Kalaiselvi, 2022*). Color changes observed ZnO-NPs formation. Nanoparticles were settled down after 20 min centrifugation, washed the nanoparticles with distilled water to remove the other impurities or extract. Zinc oxide nanoparticles were dried in an oven and stored in a refrigerator at 4 °C.

**Table 1 The representation of parameters proposed by central composite design (CCD) of response surface methodology (RSM) for the preparation of optimized ZnO-NPs.**

| Run no. | Temperature (°C) | Stirring time (min) | pH | Salt concentration (M) |
|---|---|---|---|---|
| 1 | 70 | 80 | 11 | 0.21 |
| 2 | 80 | 40 | 12 | 0.60 |
| 3 | 60 | 40 | 12 | 0.60 |
| 4 | 60 | 40 | 10 | 0.60 |
| 5 | 50 | 80 | 11 | 0.33 |
| 6 | 80 | 120 | 10 | 0.06 |
| 7 | 80 | 40 | 10 | 0.60 |
| 8 | 80 | 40 | 12 | 0.06 |
| 9 | 70 | 160 | 11 | 0.33 |
| 10 | 70 | 2.00 | 11 | 0.33 |
| 11 | 80 | 120 | 12 | 0.06 |
| 12 | 80 | 40 | 10 | 0.06 |
| 13 | 80 | 120 | 12 | 0.60 |
| 14 | 70 | 80 | 9 | 0.33 |
| 15 | 70 | 80 | 11 | 0.33 |
| 16 | 60 | 120 | 10 | 0.60 |
| 17 | 70 | 80 | 11 | 0.87 |
| 18 | 60 | 40 | 10 | 0.06 |
| 19 | 60 | 40 | 12 | 0.06 |
| 20 | 70 | 80 | 11 | 0.33 |
| 21 | 80 | 120 | 10 | 0.06 |
| 22 | 70 | 80 | 11 | 0.33 |
| 23 | 60 | 120 | 12 | 0.06 |
| 24 | 60 | 120 | 12 | 0.60 |
| 25 | 60 | 120 | 10 | 0.60 |
| 26 | 70 | 80 | 13 | 0.33 |
| 27 | 90 | 80 | 11 | 0.33 |

## Characterization of green synthesized ZnO nanoparticles

*Silybum marianum* based green synthesized ZnO-NPs were characterized by a UV-visible spectrophotometer (HITACHI U-2900) and a Zetasizer. At first, the prepared ZnO-NPs were confirmed by visual inspection. Then Zetasizer measurement was used to determine the size distribution or average size of green synthesized ZnO-NPs. The UV/ Visible spectra of ZnO-NPs were obtained by UV/visible spectrophotometer (HITACHI U-2900) with quartz cuvettes of 1.00 cm. The ZnO-NPs were visible in the region of wavelength from 250 to 378 nm.

## Pesticidal potential of green synthesized ZnO-NPs

The laboratory experiment was conducted at the Department of Entomology, University of Agriculture Faisalabad, Pakistan, to check the pesticidal activity of green synthesized

ZnO-NPs and raw seed extract of *Silybum marianum* against two pests such as *Sitophilus oryzae* (rice weevil) and *Tribolium castaneum* (red flour beetles) at various concentrations and time intervals. It was observed that only treatment ($R_{11}$) gave the nano-sized ZnO particles; that is why we only performed the pesticidal activity of this run by making different concentrations such as 0.75, 1.5, 3, and 6%. The pesticidal activity of different concentrations of raw seed extract of *Silybum marianum* and ZnO-NPs were observed in Petri plates. Twenty insects of *Sitophilus oryzae* (Taifa et al., 2022) and *Tribolium castaneum* were taken in each petri dish and covered properly after the application of the sample to check the mortality rate of both pests after the different time intervals of 24, 48, and 72 h. The mortality rate (%) was recorded by the following equation (Eq. 1):

$$Mc\ (\%) = (Mo - Me) \times 100 \qquad (1)$$

where Mc represents the actual mortality rate (%), Mo represents an observed mortality rate of pest treatments (%), and Me represents the mortality rate of control (%).

## Antimicrobial activity of ZnO-NPs

Similar to pesticidal activity, the antimicrobial activity of $R_{11}$ of ZnO-NPs and raw seed extract of *Silybum marianum* were observed. The different concentrations (1.5%, 2%, 3%) of raw seed extract of *Silybum marianum* and ZnO-NPs were used to check their antibacterial (*Pseudomonas syringae* and *Clavibacter michiganensis*) and antifungal (*Aspergillus niger* and *Fusarium oxysporum*) activities.

### Antibacterial analysis

The antibacterial activity of seed extract of *Silybum marianum* and green synthesized ZnO-NPs were observed by well diffusion method against two strains of bacteria such as *Pseudomonas syringae* (gram-negative bacteria) and *Clavibacter michiganensis* (gram-positive bacteria). For the preparation of the nutrient agar solution, 100 mL of distilled water was added into 2.3 g of nutrient agar in a conical flask, mixed the solution well, and covered with aluminium foil. After mixing the solution, it was kept in the oven at 121 °C for 15 min. Afterward, a broth solution for bacterial strains was prepared by adding 0.25 g yeast, 0.25 g NaCl, and 0.7 g tryptophan in 50 mL of distilled water. When the solution was prepared than 100 μL of these prepared bacteria, the broth was added to the flask containing the agar media with the help of micropipettes. Then this media was added onto the Petri plates and kept under laminar airflow (Rashmi Scientific Company, Chennai, India). After solidifying agar media, a well of 4 to 5 mm was made on a Petri plate using a wire loop. Then a 50 μL sample of various concentrations (1.5%, 2%, 3%) of synthesized ZnO-NPs and the seed extract was added into these wells and placed these Petri dishes into the oven overnight at 37 °C. An antibiotic such as ciprofloxacin was used as a positive control. The zone of inhibition appeared in Petri plates after 24 h and was measured with the help of a scale.

### Antifungal activity

Antifungal activity of *Silybum marianum* and their perspective ZnO nanoparticles were evaluated against two strains of fungi, including *Aspergillus niger* and *Fusarium*

*oxysporum*. At the start, the agar medium was prepared by adding 100 mL of distilled water and 4.2 g of potato dextrose agar in a conical flask and mixing well. Then keep this solution in the oven for 20 min at 120 °C. Each fungal strain was mixed with 1 mL of sterilized water. Then the agar media was put onto Petri plates. As soon as the media was presented on the Petri plates, it was solidified; each strain was uniformly swabbed onto the individual plates by using sterile cotton swabs then 4 to 5 mm wells were formed on Petri plates. A total of 50 μL of various concentrations (1.5%, 2%, and 3%) of synthesized ZnO-NPs and seed extract were added into these wells and kept plated. An antifungal tablet such as voriconazole was used as a control.

## Statistical analysis

Design-Expert software was used for the analysis and design of experimental data and to draw the 3D plots for achieving the maximum green ZnO-NPs regarding the maximum absorption and minimum particle size. ANOVA demonstrated the results of this study as mean ± standard deviation followed by Duncan's test that was carried out using Minitab software, and the level of significance ($p < 0.05$) was determined (*Shabaani et al., 2020*).

## RESULTS

### Characterization of green synthesized ZnO-NPs

Various techniques have been used to characterize the ZnO nanoparticles, such as UV visible, Zetasizer, FTIR, SEM, TEM, AFM, and XRD spectroscopy, but due to unavailability and high cost, only UV visible and Zetasizer analysis was performed in the current study. Zinc oxide nanoparticles (ZnO-NPs) formation was observed by vision when a salt solution was added to the plant extract of *Silybum marianum*. The color change from yellow to white milky confirmed the ZnO-NPs formation. *Silybum marianum* extract has a yellow color. The color change was due to the presence of the energetic molecule in *Silybum marianum* extract.

#### UV-visible spectroscopy

The maximum wavelength (λ max) for all the treatments (27) was determined using a UV-visible spectrophotometer (HITACHI U-2900) in a wavelength range from 200–600 nm. The results of UV-visible spectroscopy revealed that the absorption spectrum of ZnO-NPs was observed only for treatment ($R_{11}$) at 360.5 nm wavelength when the sample conditions were at a temperature of 80 °C, stirring time of 120 mins, pH of 12, and salt concentration of 0.06 M. Moreover, the absorption band at 360.5 nm confirmed the synthesis of ZnO-NPs through UV-visible spectroscopy (see Table 2).

#### Zetasizer measurement

A Zetasizer evaluated the average size and polydispersity of biosynthesized ZnO-NPs. The results showed the average particle size of about 51.80 nm for $R_{11}$ of ZnO-NPs at 80 °C, a stirring time of 120 min, a salt concentration of 0.06 M, and pH 12 (see Fig. 1). Results showed that particles are monodispersed in nature. The homogeneity among the size of particles was estimated by the polydispersity index (PDI). The results revealed the minimum PDI of about 0.443 for green synthesized $R_{11}$ of ZnO-NPs. Studies showed that

**Table 2 The results of response (λ max) of different runs of prepared ZnO-NPs.**

| Run no. | Temperature (°C) | Stirring time (min) | pH | Salt concentration (M) | Response (nm) λ max |
|---|---|---|---|---|---|
| 1 | 70 | 80 | 11 | 0.21 | 355 |
| 2 | 80 | 40 | 12 | 0.60 | 376 |
| 3 | 60 | 40 | 12 | 0.60 | 379 |
| 4 | 60 | 40 | 10 | 0.60 | 320 |
| 5 | 50 | 80 | 11 | 0.33 | 325 |
| 6 | 80 | 120 | 10 | 0.06 | 370 |
| 7 | 80 | 40 | 10 | 0.60 | 305 |
| 8 | 80 | 40 | 12 | 0.06 | 353 |
| 9 | 70 | 160 | 11 | 0.33 | 326 |
| 10 | 70 | 2.00 | 11 | 0.33 | 304 |
| 11 | 80 | 120 | 12 | 0.06 | 360.5 |
| 12 | 80 | 40 | 10 | 0.06 | 377 |
| 13 | 80 | 120 | 12 | 0.60 | 384 |
| 14 | 70 | 80 | 9 | 0.33 | 310 |
| 15 | 70 | 80 | 11 | 0.33 | 347 |
| 16 | 60 | 120 | 10 | 0.60 | 374 |
| 17 | 70 | 80 | 11 | 0.87 | 400 |
| 18 | 60 | 40 | 10 | 0.06 | 357 |
| 19 | 60 | 40 | 12 | 0.06 | 325.5 |
| 20 | 70 | 80 | 11 | 0.33 | 347 |
| 21 | 80 | 120 | 10 | 0.06 | 370 |
| 22 | 70 | 80 | 11 | 0.33 | 347 |
| 23 | 60 | 120 | 12 | 0.06 | 369 |
| 24 | 60 | 120 | 12 | 0.60 | 382 |
| 25 | 60 | 120 | 10 | 0.60 | 374 |
| 26 | 70 | 80 | 13 | 0.33 | 332 |
| 27 | 90 | 80 | 11 | 0.33 | 352 |

if the PDI value is less than 0.7, different size distributions are monodispersed in the sample (*Wong et al., 2020*). The above results confirmed that ZnO-NPs were monodispersed with a PDI value of 0.44, less than 0.7.

## Optimization of parameters for ZnO-NPs synthesis through response surface methodology (RSM)

Optimization by the conventional techniques was expensive, time-consuming, needed a large amount of experimental work, and involved a one-factor process. However, it did not explain the interaction effect between variables (*Garai et al., 2022*). A new statical technique was introduced called response surface methodology (RSM) to overcome this problem. It is a collection of techniques used to analyze and design various experiments and determine the effect of different variables on a response variable. The present research used the central composite design (CCD) of RSM. CCD optimized the synthesis

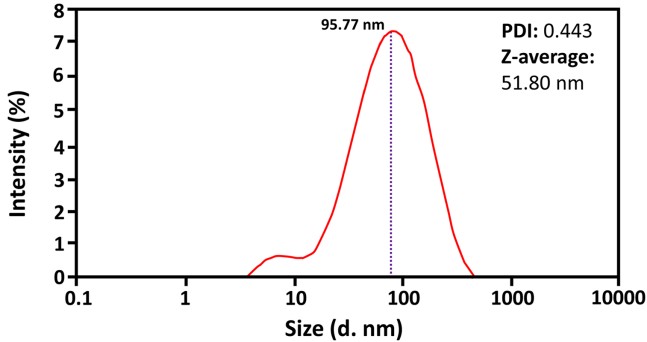

**Figure 1  The representation of results of Zetasizer of ZnO-NPs ($R_{11}$).**

parameters (temperature, pH, concentration, and stirring time) and reduced the time of experiment and cost by decreasing the number of runs performed in the laboratory (*Usman & Aziz, 2018*).

### Selection of an adequate model

The selection of the model was the first step for the optimization of synthesis parameters. Its selection was based on *p*-value and F-value. If the probability value was smaller (less than 0.05), it was significant and considered highly significant if the *p*-value was less than 0.0001.

In Table 3, a smaller *p*-value of 0.0121 and a larger F-value of 5.12 for the quadratic model indicated that the model was significant. However, the lack of fit model was non-significant for the quadratic model with a *p*-value (0.1596) and F- value (5.65). Moreover, the $R^2$ value (0.7798), adjusted $R^2$ value (0.5230), and predicted $R^2$ values (−0.5040) for the quadratic model were determined from model summary statistics. The results revealed that the quadratic model gave an outstanding description of the relationship between independent variables and response variables (wavelength). Therefore, it was selected for the optimization of a parameter to synthesize the ZnO-NPs.

### Regression analysis

The adequate model for optimizing parameters was selected based on *p* and F values. The current study used multiple regression analysis to fit the response wavelength. This model represented the response (λ max) R as a function of (A) temperature, (B) stirring time, (C) pH, and (D) salt concentration. The relation between input variables and response variables in terms of a coded factor is given below in Eq. (2):

Quadratic equation in terms of coded factors:

$$
\begin{aligned}
\text{Wavelength (response) } R = \big(&+ 340.11 + 4.13A + 10.82B + 4.33C - 2.00D \\
&- 0.88AB - 0.56AC - 3.33AD - 5.56BC + 4.94BD \\
&+ 13.25CD + 2.12A2 - 3.90B2 - 2.24C2 + 21.18\,D2\big)
\end{aligned} \tag{2}
$$

This equation explained the effect of a single variable or combination of two or more variables on response R. The Positive values of coefficients determined the positive

**Table 3 The results of analysis of variance of synthesized ZnO-NPs.**

| Source | Sum of squares | df | Mean squares | F value | p-value prob > F | Model |
|---|---|---|---|---|---|---|
| Mean *vs* total | 3.352E + 006 | 1 | 3.352E + 006 | | | Suggested |
| Linear *vs* mean | 4,163.98 | 4 | 1,041.00 | 1.45 | 0.2505 | |
| 2F1 *vs* linear | 3,887.00 | 6 | 647.83 | 0.87 | 0.5361 | |
| Quadratic *vs* 2F1 | 7,493.01 | 4 | 1,873.25 | 5.12 | 0.0121 | Suggested |
| Cubic *vs* quadra | 4,237.51 | 9 | 470.83 | 9.35 | 0.0461 | Aliased |
| Residual | 151.00 | 3 | 50.33 | | | |
| Total | 3.372E + 006 | 27 | 1.249E + 005 | | | |

interaction of variables and their effect on response R. Negative values indicated the interfering effect of input variables on response wavelength R.

The results revealed that a smaller *p*-value, such as 0.0121, and a larger F-value of 5.12 for the quadratic model indicated that the model was significant. However, the lack of fit model was non-significant for a quadratic model, confirming their maximum fitness on the effect of synthesis parameters (see Table 3). Moreover, the goodness of fit model or the coefficient of variation (C.V%) of the model was estimated by calculating the values of the coefficient of determination $R^2$, predicted $R^2$, and adjusted $R^2$. The higher $R^2$ value of 0.7798, adjusted $R^2$ value of 0.5230, and predicted $R^2$ values of −0.5040 for the quadratic model were good measures of overall accuracy and fitness (see Table 4). These results revealed that the quadratic model gave an excellent description of the relationship between independent variables (temperature, pH, stirring times, and salt concentration) and response (λ max). Therefore, it was selected to optimize conditions for the green synthesis of ZnO-NPs.

The statistical importance of the quadratic equation was determined by analysis of variance for the response surface quadratic model for (response wavelength, R). However, *p*-values were used as a significant tool for each coefficient. If it was less than 0.05, it was significant, and if it was less than 0.0001 considered highly significant. Moreover, if it was higher than 0.05, model terms were estimated as non-significant. In Table 5, the F-value of a model (3.04) and Prob > F value of 0.0305 indicated that the model was significant. The F-value for the lack and fit model was 5.65, and the Prob > F value was 0.1596, confirming that the lack and fit model was non-significant (see Table 5).

### Diagnostic plots

The analysis of the relation between predicted and actual values data was evaluated as presented in Fig. 2. From a normal plot of residual, predicted *vs* actual plot, and residual *vs* run plot, it was determined that the quadratic model was significant for estimating response over input variables. A normal probability plot of the studentized residuals checked the normality of residuals. A constant error was checked by studentized residuals *vs* predicted values. A residual *vs* run plot was used to check the outlier's values in the data.

**Table 4** The model summary of synthesized ZnO-NPs.

| Source | Std. Dev | R-squared | Adjusted R-squared | Predicted R-squared |
|--------|----------|-----------|--------------------|--------------------|
| Linear | 26.77 | 0.2089 | 0.0651 | −0.2460 |
| 2F1 | 27.25 | 0.4039 | 0.0314 | −0.8646 |
| Quadratic | 19.12 | 0.7798 | 0.5230 | −0.5040 |
| Cubic | 7.09 | 0.9924 | 0.9343 | — |

## Factor affecting the synthesis of ZnO-NPs

Various yield-affecting conditions for the synthesis of ZnO-NPs, including temperature, pH, salt concentration, and stirring time, were optimized (see Fig. 3). The absorption spectra of ZnO-NPs were recorded in the wavelength range of 250–378 nm. The results showed that a maximum amount of ZnO-NPs were synthesized at a wavelength of 360.5 nm and 80 °C. The λmax showed that when the temperature increased, the λ max was also increased, which influenced the synthesis of ZnO-NPs. Similarly, pH of the solution was another factor that influenced the synthesis of ZnO-NPs. A UV-visible study showed that basic pH (12) was favorable for forming ZnO-NPs because it abruptly changed the color of the solution from yellow to milky white. The 3D graph showed that the maximum yield of ZnO-NPs was obtained at pH 12 and a temperature of 80 °C.

In contrast, the response wavelength λ max was 360.5 nm. An increase in pH cause to enhance the nucleation center, which increased λ max and promoted the synthesis of ZnO-NPs. Furthermore, stirring time and salt concentration influenced the shape, size, and synthesis rate of ZnO-NPs. UV-visible spectroscopy showed that the synthesis of ZnO-NPs was maximum when stirring time optimized as 120 min and salt concentration as 0.06 M at λ max of 360.5 nm. At this stirring, the collision frequency of particles was high, which promoted the reduction of Zn ions into Zn metal nanoparticles.

## Pesticidal activity of ZnO-NPs

The pesticidal activity of green synthesized ZnO-NPs was evaluated against two strains of pests such as *Tribolium castaneum* and *Sitophilus oryzae*, by contact toxicity method. For this purpose, different concentrations (0.75%, 1.5%, 3% and 6%) of ZnO-NPs and raw seed extract of *Silybum marianum* were applied on both *Tribolium castaneum* and *Sitophilus oryzae* at an exposure of various times intervals such as 24, 48 and 72 h. Results revealed that the minimum mortality rate of about $9 \pm 0.57$ and $9 \pm 0.57\%$ was observed against *Tribolium castaneum* and *Silybum marianum* by 0.75 concentration of ZnO-NPs whereas seed extract showed a minimum mortality rate of $5 \pm 0.57$ and $3 \pm 0.57\%$ at the same concentration, respectively (see Table 6; Tables S1 and S2). Results of Table 6 revealed that the toxic effect of green synthesized ZnO-NPs was significant ($p < 0.05$) with a mortality rate of $78 \pm 0.57\%$ against *Tribolium castaneum* than *Silybum marianum* seeds extract ($43 \pm 0.57\%$) at 6% concentration of ZnO-NPs and seeds extract after 72 h exposure. Similarly, the pesticidal potential of ZnO-NPs was also significant ($p < 0.05$) against *Sitophilus oryzae* ($74 \pm 0.57\%$) than seeds extract of a plant that only showed the

**Table 5  Analysis of variance for interactive effect of parameters of synthesized ZnO-NPs.**

| Source | Sum of squares | df | Mean square | F-value | p-value Prob > F | Model |
|---|---|---|---|---|---|---|
| Model | 1,554,399 | 14 | 1,110.29 | 3.04 | 0.0305 | Significant |
| A-Temperature | 408.38 | 1 | 408.38 | 1.12 | 0.3114 | |
| B-Stirring time | 2,783.74 | 1 | 2,783.74 | 7.61 | 0.0173 | |
| C-Ph | 450.67 | 1 | 450.67 | 1.23 | 0.2887 | |
| D-Salt concentration | 69.37 | 1 | 69.37 | 0.19 | 0.6709 | |
| AB | 12.25 | 1 | 12.25 | 0.033 | 0.8578 | |
| AC | 5.06 | 1 | 5.06 | 0.014 | 0.9083 | |
| AD | 175.56 | 1 | 175.56 | 0.48 | 0.5016 | |
| BC | 495.06 | 1 | 495.06 | 1.35 | 0.2672 | |
| BD | 390.06 | 1 | 390.06 | 1.07 | 0.3221 | |
| CD | 2,809.00 | 1 | 2,809.00 | 7.68 | 0.0169 | |
| A^2 | 107.48 | 1 | 107.48 | 0.29 | 0.5977 | |
| B^2 | 341.30 | 1 | 341.30 | 0.93 | 0.3531 | |
| C^2 | 117.33 | 1 | 117.33 | 0.32 | 0.5816 | |
| D^2 | 6,397.34 | 1 | 6,397.34 | 17.49 | 0.0013 | |
| Residual | 4,388.51 | 12 | 365.71 | | | |
| Lack of fit | 4,238.511 | 10 | 423.85 | 5.65 | 0.1596 | Non-significant |
| Pure error | 150.00 | 2 | 75.00 | — | — | |
| CCR total | 1,9932.50 | 26 | — | — | — | |

mortality rate of 25 ± 1.52% at 6% concentration after 72 h observation (see Figs. S1 and S2).

Moreover, when comparing these results with a positive standard such as Pyriproxyfen, it was noted that the mortality rate increased to 100 ± 0.0 after 72 h exposure at 6% concentration. However, it was noticed that the effect of ZnO-NPs was concentration and exposure-time dependent. When the concentration was increased, the mortality rate of pests was also enhanced. However, it is concluded that the synthesized ZnO-NPs showed better pesticidal effects than the raw seed extract of *Silybum marianum*.

## Antimicrobial activity of ZnO-NPs
### Antibacterial activity of ZnO-NPs

The antibacterial potential of ZnO-NPs and *S. marianum* seeds extract was assessed against two bacterial strains, *Clavibacter michigannessis*, and *Pseudomonas syringea*, by agar well diffusion method. Different concentrations, such as 0.75%, 1.5%, and 3% of green synthesized ZnO-NPs and seeds extract of the plant, were applied on both bacterial strains at a time interval of 24 h. Ciprofloxacin was used as a positive control.

The results revealed that the maximum zone of inhibition of about 18 ± 0.4 and 17 ± 0.25 mm was observed against *Clavibacter michigannessis* at a 3% concentration of ZnO-NPs and seed extract of *Silybum marianum*, respectively (see Table 7; Fig. S3, Tables S3 and S4 in supplemental file). In contrast, the maximum zone of inhibition

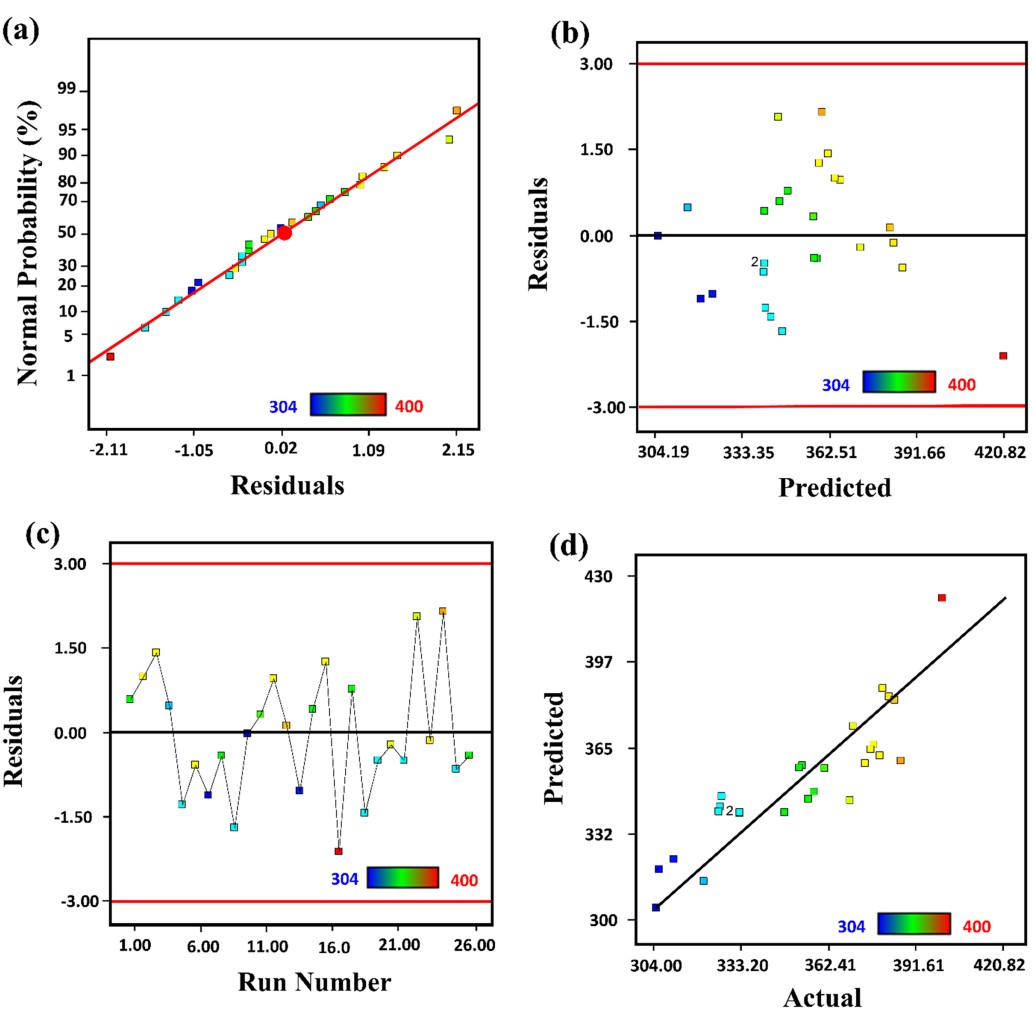

**Figure 2** The representation of (A) normal plot of residual, (B) predicted *vs* actual, (C) residual *vs* run, and (D) residual *vs* predicted plot for the synthesized of ZnO-NPs by *Silybum marianum*.

of about 25 ± 0.4 and 20 ± 0.4 mm was noted against *Pseudomonas syringae* at 3% concentration of ZnO-NPs and plant seed extract, respectively (see Table 7; Fig. S4, Tables S5 and S6 in supplemental file). The inhibitory effect of ZnO-NPs (25 ± 0.4 mm) was comparable with ciprofloxacin (35 ± 0.65 mm). Results reported that ZnO-NPs showed better antibacterial activity against *Pseudomonas syringae* than *Clavibacter michigannessis*. It was observed that when concentration decreased, the zone of inhibition also reduced. Therefore, ZnO-NPs showed a greater effect on the zone of inhibition than the seed extract of the plant. The results confirmed that the antibacterial activity of ZnO-NPS was also concentration-dependent when higher concentrations (3%) were used; the inhibitory effect of ZnO-NPS and seeds extract of *Silybum marianum* was greater and vice versa.

### Antifungal activity of ZnO-NPs

ZnO-NPs have the ability to reduce the growth of phytopathogenic fungi. Antifungal activity of ZnO-NPs and seed extract of *Silybum marianum* was assessed against

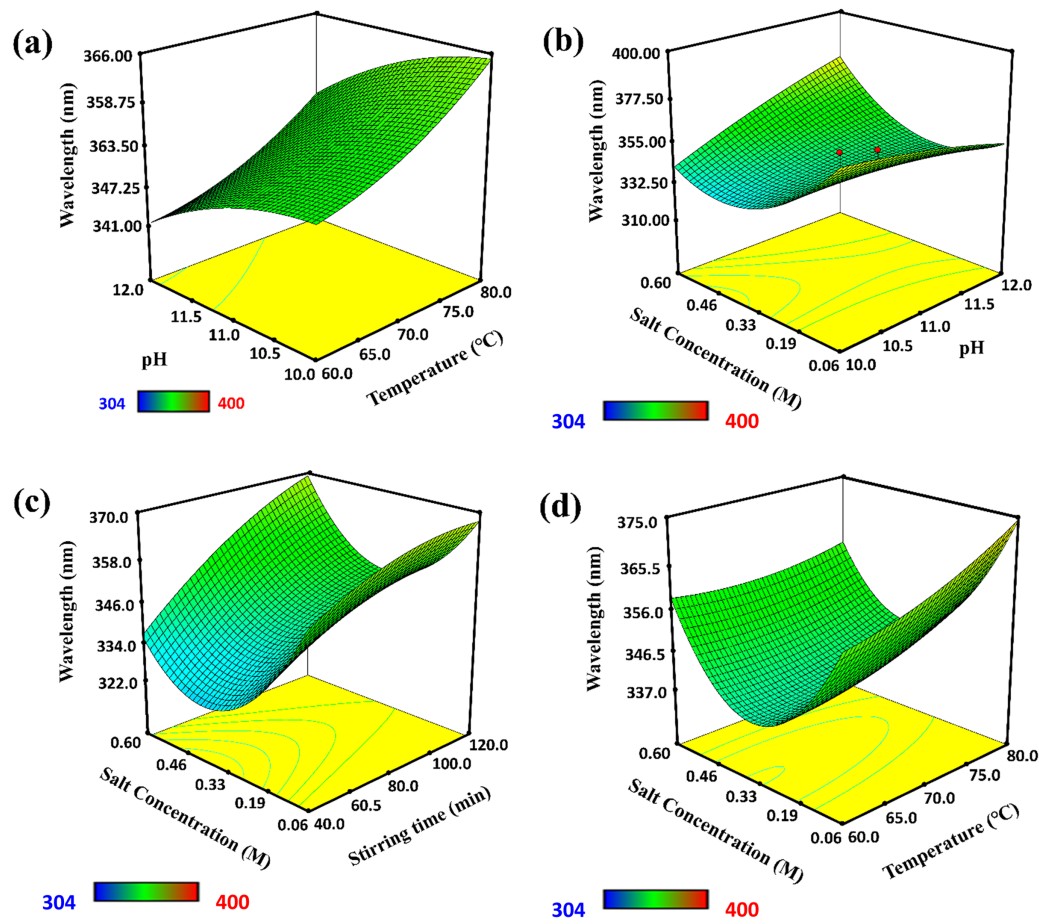

**Figure 3 3D graphs of factors affecting the synthesis of ZnO-NPs.** (A) The effect of temperature, (B) the effect of pH, (C) effect of stirring time, and (D) effect of salt concentration.

*Aspergillus niger* and *Fusarium oxysporum* by agar well diffusion method (*Patwekar et al., 2010*). To evaluate the antifungal inhibitory effect of green synthesized ZnO-NPs, different concentrations of 0.75%, 1.5%, and 3% of ZnO-NPs, *Silybum marianum* seeds extract, and voriconazole (control) were used.

The results showed that the maximum antifungal potential of ZnO-NPs was observed against *Aspergillus niger* (19 ± 0.8 mm) than seeds extract of *Silybum marianum* (18 ± 0.6 mm) at a sample concentration of 3% (see Table 8; Fig. S5, Tables S7 and S8 in supplemental file). Similarly, the inhibitory effect of green synthesized ZnO-NPs was also significant ($p < 0.05$) against *Fusarium oxysporum*, which showed the maximum zone of inhibition of 21 ± 0.57 mm at 3% concentration (see Table 8; Fig. S6, Tables S9 and S10 in supplemental file). At a lower concentration of 0.75% or 1.5%, the inhibitory effect of ZnO-NPs was less owing to less availability of active centers in the pathogenic genome. Voriconazole was used as a positive standard, and its results revealed the maximum zone of inhibition of 30.1 ± 0.50 and 35.1 ± 0.50 mm against *Aspergillus niger* and *Fusarium oxysporum*, respectively. The results confirmed that the antifungal potential of green synthesized ZnO-NPs was concentration-dependent. Overall, it is concluded that

**Table 6  The pesticidal activity of green synthesized ZnO-NPs and raw seed extract of *Silybum marianum* at different concentrations.**

| Sample | Concentration (%) | Average mortality (%) | | | | | |
|---|---|---|---|---|---|---|---|
| | | *Tribolium castaneum* | | | *Sitophilus oryzae* | | |
| | | 24 | 48 | 72 | 24 | 48 | 72 |
| ZnO-NPs | 0.75 | 9 ± 0.57[Cgh] | 27 ± 1.00[Bgh] | 43 ± 1.15[Agh] | 9 ± 0.57[Cfg] | 25 ± 1.15[Bfg] | 34 ± 1.00[Afg] |
| | 1.5 | 18 ± 0.57[Cfg] | 34 ± 1.00[Bfg] | 56 ± 0.57[Afg] | 16 ± 0.57[Cef] | 29 ± 0.57[Bef] | 43 ± 1.00[Aef] |
| | 3 | 20 ± 1.00[Cef] | 40 ± 1.00[Bef] | 69 ± 1.57[Aef] | 20 ± 1.00[Cde] | 36 ± 0.57[Bde] | 58 ± 1.15[Ade] |
| | 6 | 25 ± 0.57[Ce] | 45 ± 0.57[Be] | 78 ± 0.57[Ae] | 23 ± 1.52[Cd] | 38 ± 0.57[Bd] | 74 ± 0.57[Ad] |
| *Silybum marianum* seed extract | 0.75 | 5 ± 0.57[Cj] | 7 ± 1.00[Bj] | 9 ± 0.57[Aj] | 3 ± 0.57[Ci] | 5 ± 1.00[Bi] | 7 ± 0.57[Ai] |
| | 1.5 | 9 ± 0.57[Cij] | 16 ± 0.57[Bij] | 23 ± 0.57[Aij] | 5 ± 0.57[Chi] | 7 ± 0.57[Bhi] | 12 ± 1.00[Ahi] |
| | 3 | 12 ± 0.57[Chi] | 20 ± 1.00[Bhi] | 28 ± 1.15[Ahi] | 9 ± 0.57[Cgh] | 16 ± 0.57[Bgh] | 23 ± 0.57[Agh] |
| | 6 | 18 ± 0.57[Cgh] | 27 ± 1.00[Bgh] | 43 ± 0.57[Agh] | 16 ± 1.00[Cfgh] | 18 ± 0.57[Bfgh] | 25 ± 1.52[Afgh] |
| Synthetic drug (pyriproxyfen) | 0.75 | 33.3 ± 1[Cd] | 62.22 ± 1.52[Bd] | 88.8 ± 0.57[Ad] | 37.7 ± 1.52[Cc] | 64.4 ± 2.00[Bc] | 91.1 ± 0.57[Ac] |
| | 1.5 | 51.1 ± 2.5[Cc] | 77.7 ± 1.52[Bc] | 93.3 ± 1.0[Ac] | 55.5 ± 1.52[Cb] | 80 ±1.00[Bb] | 95.5 ± 1.15[Ab] |
| | 3 | 71.1 ± 1.52[Cb] | 84.4 ± 0.57[Bb] | 97.7 ± 0.57[Ab] | 75.5 ± 1.52[Cb] | 86.6 ± 0.00[Bb] | 100 ± 0.00[Ab] |
| | 6 | 97.7 ± 0.57[Ca] | 100 ± 0.0[Ba] | 100 ± 0.00[Aa] | 100 ± 0.00[Ca] | 100 ± 0.00[Ba] | 100 ± 0.00[Aa] |

Note:
Values are expressed as mean ±SD indicating the % average mortality. Within each column, means followed by the same uppercase letter are not significantly different, within each row means followed by the same lowercase letter are not significantly different; Tukey HSD test; $p < 0.05$.

**Table 7  Anti-bacterial activity of ZnO-NPs and raw seed extract of *Silybum marianum* at different concentrations.**

| Sample | Average zone of inhibition (mm) | | | | | |
|---|---|---|---|---|---|---|
| | *C lavibacter michiganensis* | | | *Pseudomonas syringae* | | |
| | 0.75 | 1.5 | 3 | 0.75 | 1.5 | 3 |
| ZnO-NPs | 8.0 ± 0.40[Bc] | 14 ± 0.35[Bb] | 18 ± 0.40[Ba] | 10 ± 0.40[Bc] | 19 ± 0.30[Bb] | 25 ± 0.40[Ba] |
| *Silybum marianum* seed extract | 6.0 ± 0.35[Cc] | 12 ± 0.35[Cb] | 17 ± 0.25[Ca] | 8.0 ± 0.35[Cc] | 15 ± 0.35[Cb] | 20 ± 0.25[Ca] |
| Ciprofloxacin (control) | 19.5 ± 0.35[Ac] | 26.8 ± 0.35[Ab] | 33.6 ± 0.25[Aa] | 20.3 ± 0.40[Ac] | 28.8 ± 0.30[Ab] | 35.4 ± 0.40[Aa] |

Note:
Values are expressed as mean ±SD indicating the average zone of inhibition. Within each row, means followed by the same uppercase letter are not significantly different, within the column means followed by the same lowercase letter are not significantly different; Tukey HSD test; $p < 0.05$).

ZnO-NPs observed maximum inhibitory zones than seeds extract of the plant at 3% while it was less than control.

# DISCUSSION

Globally, each year grain yield is wasted during storage due to insects and pests (*Rani et al., 2021*). *Tribolium castaneum* and *Sitophilus oryzae* are destructive wheat, rice, and maize pests worldwide. *Clavibacter michiganensis* and *Pseudomonas syringes* are gram-positive and gram-negative phytopathogenic bacterial strains responsible for canker (*Atiq et al., 2022*) and wilting in tomatoes, ring rot in potatoes, and bacterial apical necrosis (BAN) in mango trees (*Miller et al., 2019*). Similarly, *Fusarium oxysporum* and *Aspergillus niger* also cause black mold disease in vegetables and fruits such as apricots, onions, grapes, and peanuts (*Tolossa & Shibeshi, 2022*). These pests have majorly been managed using synthetic pesticides. However, many serious concerns are associated with using chemical

**Table 8 Antifungal potential of ZnO-NPs and raw seed extract of *Silybum marianum* at different concentrations.**

| Sample concentration (%) | Average zone of inhibition (mm) | | | | | |
|---|---|---|---|---|---|---|
| | *Aspergillus niger* | | | *Fusarium oxysporum* | | |
| | 0.75 | 1.5 | 3 | 0.75 | 1.5 | 3 |
| ZnO nanoparticles | 10 ± 0.80[Bc] | 12 ± 0.60[Bb] | 19 ± 0.80[Ba] | 13 ± 0.71[Bc] | 16 ± 0.60[Bb] | 21 ± 0.57[Ba] |
| *Silybum marianum* seed extract | 7.0 ± 0.20[Cc] | 11 ± 0.00[4Cb] | 18 ± 0.60[Ca] | 12 ± 0.20[Cc] | 15 ± 0.71[Cb] | 18 ± 0.40[Ca] |
| Voriconazole (control) | 15.2 ± 0.40[Ac] | 27.3 ± 0.40[Ab] | 30.1 ± 0.50[Aa] | 17.4 ± 0.66[Ac] | 30.8 ± 0.23[Ab] | 35 ± 0.65[Aa] |

**Note:**
Values are expressed as mean ±SD indicating the average zone of inhibition. Within each row, means followed by the same uppercase letter are not significantly different, within each column, means followed by the same lowercase letter are not significantly different; Tukey HSD test; $p < 0.05$.

pesticides, including harmful effects on non-target species and the environment, non-biodegradable nature, pest resistance, and human health safety. It was estimated that many people get poisoned by pesticides in a year due to inappropriate use, and 75% of these are agricultural workers (*Singh, Kaur & Aggarwal, 2022*). Based on these facts, there is a considerable demand for safe and environmentally friendly pesticides. Therefore, researchers are trying to develop biocompatible, sustainable, and nontoxic biopesticides that have encouraging applications in different fields of agriculture and medicine (*Palermo et al., 2021*). In this regard, ZnO-NPs have proved to be a promising source of safe insecticides, very effective in controlling store grain pests and phytopathogenic bacterial and fungal strains at environmentally friendly and very low dosages (*Jasrotia et al., 2022*). This study aimed to synthesize zinc oxide nanoparticles (ZnO-NPs) with seeds extract of *Silybum marianum*, which have been used as novel green pesticides to replace present synthetic pesticides. Green synthesized ZnO-NPs has unique properties such as small size, high surface area to volume ratio, biodegradability, less toxicity and remarkable stability that enhances their use in drug delivery as an antifungal, antibacterial, and pesticidal agent.

The temperature was the most significant factor for the synthesis of nanoparticles, which controls the rate of growth and size of nanoparticles by affecting nucleation. This study optimized the temperature as 80 °C at λ max of 360.5 nm. *Karam & Abdulrahman (2022)* reported the synthesis of ZnO nanoparticles from the leaf extract of a thyme plant by using a green method at different temperatures of 150 °C, 250 °C, 350 °C, and 450 °C. Similarly, pH was optimized as 12. At this pH, maximum ZnO-NPs were synthesized.

Further, increase in pH to 13, the aggregation and deformation of biomolecules occurred, which decreased the response wavelength and is responsible for forming larger ZnO-NPs. *Shabaani et al. (2020)* reported the synthesis of ZnO-NPs using plant extracts of *Eriobotrya japonica* and *Musa acuminata* at various pH values. They found that basic pH 12 was important for the production of ZnO-NPs. Similarly, *Umamaheswari et al. (2021)* synthesized ZnO-NPs from the *Raphanus sativus* extract at pH 14,12, 10, and 8. They observed that no absorption peak was reported at pH 8-10 or 14, but a characteristic absorption peak of 369 nm was found at pH value of 12.

Stirring time and salt concentrations were other factors that influenced the shape, size, and synthesis rate of ZnO-NPs. The optimized stirring time was 120 min, and the salt concentration of 0.06 M for the maximum synthesis of ZnO-NPs. At this stirring, the

collision frequency of particles was high, which promoted the reduction of Zn ions to Zn metal particles. *Manzhi et al. (2019)* reported that the variation in time occurred in different ways, such as aggregation and shrinkage of particles due to long storage time. It also affected the crystal structure and shape of ZnO-NPs. *Rasli, Basri & Harun (2020)* observed the effect of salt concentration on the morphology of ZnO-NPs synthesized using aloe vera extract. *Abdullah et al. (2020)* reported the synthesis of ZnO-NPs using peel extract of *Musa acuminata* at 0.01 M/L concentration. It was found that maximum nanoparticles were formed at the lowest salt concentration. Moreover, *Jamdagni, Khatri & Rana (2018)* synthesized ZnO-NPs using plant extract of *Cassia auriculata* at different reaction times 0.5, 1, and 2 h. It was examined from the results that a higher yield of ZnO-NPs was formed when the reaction time was 2 h.

Zetasizer analysis of green synthesized ZnO-NPs revealed that particles were monodispersed with a minimum size of 51.80 nm. Nano size provides a larger surface area for actions and greater availability of ions that improve the activity of ZnO-NPs. *Kamarajan et al. (2022)* reported that *Acalypha indica* extract-based ZnO-NPs showed a particle size of 46 nm at different conditions. It was found that these nanoparticles were monodispersed and showed maximum (96%) degradation efficiency.

Results of the pesticide activity of ZnO-NPs revealed that maximum pesticide activity was observed at 6% concentration of ZnO-NPs and seed extract of *Silybum marianum* after 72 h against both pests as *Tribolium castaneum* and *Sitophilus oryzae*. Our results are in accordance with the result of *Haider et al. (2020)*, who found that a toxicity bioassay of *Tribolium castaneum* was carried out by the plant extracts of three different concentrations (5%, 10%, and 15%) after a time interval of 24, 48 and 72. They reported the minimum mortality rate of pests of about 15.10% and the highest mortality rate of 46.12% against *Tribolium castaneum*. However, in the case of nanoparticles, the maximum mortality rate of 66.32% was noticed against *Tribolium castaneum*.

Moreover, the study was conducted on the therapeutic potential of CuNPs for curing bovine mastitis. For this purpose, various groups of rats were taken, and 6.25 g/mL (25 nm) CuNPs were chosen as the intramammary (IM) treatment for *S. aureus*-induced mastitis in rats since the *in vitro* sensitivity test revealed a significant zone of inhibition at this dose and negligible cell damage on fibroblast cell lines (*Taifa et al., 2022*). Similarly, vancomycin-based silver nanoparticles were synthesized as nano-drug complexes (van@AgNPs), and their synergetic antibacterial efficacy against *Staphylococcus aureus* and *Escherichia coli* was evaluated by Well Diffusion Method. The antibacterial potential of both bacteria was considerably increased. The present study revealed that ZnO-NPs showed better pesticide results than seed extract of *Silybum marianum*.

The results of antibacterial activity demonstrated that ZnO-NPs exhibited higher antibacterial activity against *Clavibacter michigannessis* (18 ± 0.4 mm) and *Pseudomonas syringes* (25 ± 0.4 mm) at a sample concentration of 3%. It was estimated that increased ZnO-NPs concentration enhanced the release of reducing sugar, proteins, and DNA from the extracellular cytosol of the cell membrane. It also increased the reactive oxygen species (ROS) that destroyed the bacterial cells owing to these showing greater zone of inhibition and greater antibacterial activity. Moreover, the above results clarified that ZnO-NPs had

greater antimicrobial activity ($p < 0.001$) against gram-negative bacteria (*Pseudomonas syringae*) as compared to Gram-positive bacteria (*Clavibacter michiganensis*). The reason is that gram-negative bacteria have an outer membrane that changes the hydrophobic properties or causes mutations. On the other hand, Gram-positive bacteria lack this important membrane, making them vulnerable. *Obeizi et al. (2020)* found that ZnO-NPs synthesized from the essential oil of *Eucalyptus globulus* exhibited a maximum inhibitory effect of 19.35 ± 0.45 mm against *Stymphilococus pneumoniae* at a concentration of 100 µg/mL. Moreover, the SeNPs were synthesized from the leaf extract of *Elaeagnus indica*, and their antimicrobial potential was evaluated. These green synthesized nanoparticles (10–15nm) showed significant antibacterial potential against the strains of *Salmonella typhimurium* and *Fusarium oxysporum*. The concentration (10 µg/mL) was noted as the lowest minimum inhibitory concentration against both pathogens (*Indhira et al., 2023*).

Furthermore, antifungal activity results revealed that ZnO-NPs showed more activity than seed extract. The zone of inhibition for *Aspergillus niger* was noted as 19 ± 0.8 and 21 ± 0.57 mm against *Fusarium oxysporum* at a sample concentration of 3%. The results of the present study showed that the antifungal activity of ZnO-NPs was concentration-dependent. The maximum inhibitory effect of ZnO-NPs was conducted at a higher concentration of 3% for both strains. *Madan, Wasewar & Kumar, 2017* reported that the availability of ions increased due to higher concentration. More $Zn^{+2}$ ions attached to pathogen biomolecules caused DNA destruction and produced reactive oxygen species and oxidative stress in the fungal cell membrane. However, due to the high availability of (H and O) atoms, the synthetic drug voriconazole showed a greater inhibition zone than seed extract and ZnO-NPs. *Horue et al. (2020)* found that ZnO-NPs synthesized from *Pongamia glabra* extract exhibited a maximum inhibitory effect of 18 ± 0.6 mm against *Rhizopus nigricans*.

Moreover, *Nassar (2018)* reported that *Azadirachta indica* based ZnO-NPs exhibited a maximum zone of inhibition of 17.80 ± 4.52 mm against *Sclerotium rolfsii* and *Rhizoctonia solani*. However, there are some limitations of our study concerned with the complexity of plant components, their seasonal and geographical distribution, and low purity and yield. Several plant materials are available for the green synthesis of NPs, and different investigators have explored locally present and abundantly available plants. Such studies provide the opportunity to fully use native plants, but large-scale worldwide production of green-synthesized nanoscale metals is difficult to achieve. The plant synthesizing silver nanoparticles is found in India, the Philippines, Sri Lanka, Malaysia, and South China (*Roopan et al., 2019*). Au-NPs were synthesized from Fenugreek, which was widely distributed on the east coast of the Mediterranean and China, while peppermint is native to West Asia and Central Europe (*Aromal & Philip, 2012*). Similarly, the synthesis process concerns long reaction time, use of other industrial chemical reagents, and excessive energy consumption. For example, Cu NPs were synthesized at 80 °C for 2 h (*Muthuvel, Jothibas & Manoharan, 2020*), and Au NPs were synthesized from *Cystoseira baccata* (brown algae) at −24 °C, which also consumed a large amount of energy and intensive equipment such that freezer (*Khatami et al., 2018*).

## CONCLUSIONS

Green ZnO-based NPs of *Silybum marianum* exhibited a promising potential to be used as a substitute for the more harmful insecticides. Although the environmental effects of using ZnO nanoparticles as an insecticide should be investigated further, one obvious advantage of using them as insecticides is the low risk of insect resistance developing over the course of time. Additionally, Zn is one of the micronutrients found in both human and animal diets, so when consumed by either of them, it tends to be beneficial rather than harmful. The results of this research demonstrate the potential for using ZnO nanoparticles to not only remove *Tribolium castaneum* and *Sitophilus oryzae* but also significantly decrease their population in the ecosystem by causing deformations, reduce oviposition, reduce fecundity, and egg hatchability. It is a useful tool for integrated pest management. Moreover, it is suggested that *Silybum marianum* plant could be used in combination with another plant and would show more efficient activity for other pests like an aphid.

### Funding

This research has been supported by the Endowment Fund Secretariat University of Agriculture, Faisalabad, Pakistan, grant No. RD-028-18 and also supported by the Researchers Supporting Project number (RSP2023R194), King Saud University, Riyadh, Saudi Arabia. The funders had no role in study design, data collection and analysis, decision to publish, or preparation of the manuscript.

### Grant Disclosures

The following grant information was disclosed by the authors:
Endowment Fund Secretariat University of Agriculture: No. RD-028-18.
Researchers Supporting Project: No. RSP2023R194.
King Saud University, Riyadh, Saudi Arabia.

### Competing Interests

The authors declare that they have no competing interests.

### Author Contributions

- Nazish Jahan conceived and designed the experiments, performed the experiments, analyzed the data, prepared figures and/or tables, and approved the final draft.
- Kousar Rasheed conceived and designed the experiments, performed the experiments, analyzed the data, prepared figures and/or tables, and approved the final draft.
- Khalil-Ur- Rahman conceived and designed the experiments, analyzed the data, prepared figures and/or tables, and approved the final draft.
- Abu Hazafa conceived and designed the experiments, analyzed the data, prepared figures and/or tables, and approved the final draft.
- Amna Saleem performed the experiments, prepared figures and/or tables, and approved the final draft.

- Saud Alamri analyzed the data, prepared figures and/or tables, authored or reviewed drafts of the article, revised the manuscript, funding, and approved the final draft.
- Muhammad Omer Iqbal analyzed the data, prepared figures and/or tables, authored or reviewed drafts of the article, and approved the final draft.
- Md Atikur Rahman analyzed the data, prepared figures and/or tables, authored or reviewed drafts of the article, revised the manuscript, funding, and approved the final draft.

## Data Availability

The raw measurements are available in the Supplementary Files.

## Supplemental Information

Supplemental information for this article can be found online at http://dx.doi.org/10.7717/peerj.15743#supplemental-information.

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
