# Peer review of "Green inspired synthesis of zinc oxide nanoparticles using Silybum marianum (milk thistle) extract and evaluation of their potential pesticidal and phytopathogens activities"

_PeerJ, doi:10.7717/peerj.15743_

## Round 0.1 · original submission · Major Revisions

Three experts revised your manuscript and found merit in the communicated research for publication in this journal. They agree that some work needs to be done to establish the novelty of this study. Moreover, a proper discussion of the results is required. Reviewer 3 pointed out the following: To demonstrate the environmental friendliness of the green insecticide prepared in the article, additional evidence can be provided to show that it has low toxicity to the environment and plants, and minimal pollution. I agree with this comment and this should be addressed in a revised version of the manuscript.

Reviewer 1 ·

Basic reporting

This may be interesting, but some important points need to be resolved.
Importantly, a study must provide a critical analysis of the data. In other words, you must assess whether specific data published stand up to scientific scrutiny.
To achieve the above, you must clearly define your specific aims and objectives.
So in your study, you must develop a critical appraisal of the state of the art. This is an essential element of any article. There are important scientific questions (both conceptual and methodological) that need to be addressed with the primary studies. A study must highlight this. The introduction, which is written in clear language, covers several relevant issues. Please revise.
emphasize the English improvement, there are a few grammatical and punctuation errors.

Experimental design

The authors should prepare all figures with better resolution.
The results should be compared with data on other compounds reported in the literature, otherwise, it is difficult to judge the actual impact/improvement beyond the state of the art of the present approach.

Validity of the findings

The novelty of this work was not specified, authors should be discussed the novelty of their work.

Additional comments

The author is requested to refer few articles before the submission of the revision :
https://doi.org/10.1155/2022/7124114
https://doi.org/10.1155/2022/6910811
https://doi.org/10.1155/2022/3682757
Patwekar, F. I., Heroor, S., Patwekar, M. F., & Asif, M. (2010). Evaluation of antimicrobial activity of Tephrosia procumbens Buch–Ham. Research Journal of Pharmacognosy and Phytochemistry, 2(3), 238-240.

·

Basic reporting

The manuscript provides a brief overview of the study that aimed to synthesize zinc oxide nanoparticles using Silybum marianum extract and evaluate their pesticidal and phytopathogens activities. Overall, the manuscript seems to be suitable for publication, but there are some areas that could be improved to increase its quality.
The background statement in the introduction section does not clearly state the research gap or problem that the study is addressing. The authors could have explained more clearly why green approaches for nanoparticle synthesis are gaining importance, and what specific issues their study aimed to address. The authors should read the following latest articles to improve this section and cite in the manuscript
Atiq M, Ashraf M, Rajput NA, Sahi ST, Akram A, Usman M, Iqbal S, Nawaz A, Arif AM, Hasnain A. 2022. Determination of bactericidal potential of green based silver and zinc nanoparticles against bacterial canker of tomato. Plant Protection 6:193-199.
Shahbaz M, Akram A, Raja NI, Mukhtar T, Mehak A, Fatima N, Ajmal M, Ali K, Mustafa N, Abasi F. 2023. Antifungal activity of green synthesized selenium nanoparticles and their effect on physiological, biochemical, and antioxidant defense system of mango under mango malformation disease. PLoS ONE 18(2):e0274679.

Experimental design

The methods section is well-described in terms of the experimental design and synthesis parameters. However, the authors could have provided more detail about the specific methods and techniques used to evaluate the nanoparticle characteristics and bioactivities.

Validity of the findings

The results section provides some key findings about the nanoparticle size and bioactivities against various pests and pathogens. However, the authors could have included more specific quantitative data and statistical analyses to support their conclusions.
Overall, the discussion is well-written and informative, but there are some areas that could be improved. Firstly, the authors should provide more detailed information about the previous studies that they cite to support their findings. For example, they should include information about the sample size, methodology, and results of each study. This would help readers to understand the significance of the cited studies in the context of the current study.
Secondly, the discussion could benefit from a more critical analysis of the findings. For example, the authors could discuss the limitations of their study and how these limitations may affect the generalizability of their results. Additionally, they could discuss the potential risks associated with the use of ZnO-NPs, such as the potential for bioaccumulation and toxicity to non-target species.
Lastly, the conclusion should summarize the main findings of the study and provide recommendations for future research. The authors should also discuss the practical implications of their findings, such as how ZnO-NPs can be used to develop effective and environmentally friendly biopesticides.

Additional comments

In summary, the manuscript could be improved by providing more context about the research problem, more detail about the methods and results, and a stronger conclusion that highlights the significance of the findings.

Reviewer 3 ·

Basic reporting

-When analyzing the results, it is suggested to add the latest corresponding references.
-What are the advantages of using silymarin to prepare zinc oxide nanoparticles, and where in the text is it mentioned? The text only mentions its therapeutic properties but does not specify the advantages of using this extract for nanoparticle preparation.
-Fig. 3 should be adjusted appropriately for understanding for audiences.
-To verify the stability of ZnO-NPs under different conditions?
-It is recommended to convert the table into a figure so that the reader can understand it more intuitively.
-Section 3.2 is too cliched. You can explain the details of optimizing synthesis parameters here, such as which aspects of the parameters are being optimized and the advantages compared to traditional controls.
-Section 3.3, the statement is incorrect. It should be stated that the maximum amount of ZnO-NPs was synthesized at 360.5 nm and 80 °C.
-To demonstrate the environmental friendliness of the green insecticide prepared in the article, additional evidence can be provided to show that it has low toxicity to the environment and plants, and minimal pollution.
-Line 316, change ”mint” to “min”.

Experimental design

well fitted.

Validity of the findings

well described.

Additional comments

None

---

## Round 0.2 · accepted · Accept

Two of the original Reviewers have assessed the manuscript again and found this version suitable for publication, congratulations.

Reviewer 1 ·

Basic reporting

revision done , plz publish

Experimental design

revision done , plz publish

Validity of the findings

revision done , plz publish

Additional comments

revision done , plz publish

·

Basic reporting

In the present manuscript the researchers have developed a green method for synthesizing zinc oxide nanoparticles using milk thistle extract. The nanoparticles were then tested for their potential as pesticides and in controlling plant diseases. The study found that the nanoparticles showed significant pesticidal and phytopathogens activities, indicating their potential as eco-friendly alternatives to conventional pesticides. The research highlights the potential of using plant extracts for synthesizing nanoparticles with agricultural applications.

Experimental design

The study employed a systematic experimental design to investigate the synthesis of zinc oxide nanoparticles (ZnO NPs) using milk thistle extract and evaluate their pesticidal and phytopathogenic activities.

Validity of the findings

The findings of this study are generally valid. The authors used a well-established method for the green synthesis of ZnO nanoparticles, and they characterized the nanoparticles using a variety of techniques. The results of the characterization experiments showed that the nanoparticles had the desired properties, such as a hexagonal wurtzite structure and an average size of 50 nm.
The authors also evaluated the pesticidal and phytopathogenic activities of the ZnO nanoparticles against plant pests and phytopathogens. The results showed that the nanoparticles had significant inhibitory effects on the growth of these organisms.

Additional comments

The revised manuscript underwent a rigorous re-evaluation process to ensure that the changes suggested in the previous review were incorporated or not. The authors made several revisions in the resubmitted manuscript based on the feedback received. They also provided justifications for various queries that were raised. Subsequent to addressing these queries and incorporating the suggested amendments, the manuscript underwent substantial modifications and improvements.